# Specific Recognition of the 5′-Untranslated Region of West Nile Virus Genome by Human Innate Immune System

**DOI:** 10.3390/v14061282

**Published:** 2022-06-13

**Authors:** Emmanuelle Bignon, Marco Marazzi, Tom Miclot, Giampaolo Barone, Antonio Monari

**Affiliations:** 1Université de Lorraine and CNRS, LPCT UMR 7019, F-54000 Nancy, France; tom.miclot@unipa.it; 2Grupo de Reactividad y Estructura Molecular (RESMOL), Departamento de Química Analítica, Química Física e Ingeniería Química, Universidad de Alcalá, Alcalá de Henares, 28805 Madrid, Spain; marco.marazzi@uah.es; 3Instituto de Investigación Química “Andrés M. del Río” (IQAR), Universidad de Alcalá, Alcalá de Henares, 28805 Madrid, Spain; 4Department of Biological, Chemical and Pharmaceutical Sciences, Università degli Studi di Palermo, viale delle Scienze, 90128 Palermo, Italy; giampaolo.barone@unipa.it; 5ITODYS, Université Paris Cité, CNRS, F-75006 Paris, France

**Keywords:** West Nile Virus, oligoadenylate synthetase 1, 5′-untranslated region, recognition mechanism, immune system, emerging viruses

## Abstract

In the last few years, the sudden outbreak of COVID-19 caused by SARS-CoV-2 proved the crucial importance of understanding how emerging viruses work and proliferate, in order to avoid the repetition of such a dramatic sanitary situation with unprecedented social and economic costs. West Nile Virus is a mosquito-borne pathogen that can spread to humans and induce severe neurological problems. This RNA virus caused recent remarkable outbreaks, notably in Europe, highlighting the need to investigate the molecular mechanisms of its infection process in order to design and propose efficient antivirals. Here, we resort to all-atom Molecular Dynamics simulations to characterize the structure of the 5${}^{\prime }$-untranslated region of the West Nile Virus genome and its specific recognition by the human innate immune system via oligoadenylate synthetase. Our simulations allowed us to map the interaction network between the viral RNA and the host protein, which drives its specific recognition and triggers the host immune response. These results may provide fundamental knowledge that can assist further antivirals’ design, including therapeutic RNA strategies.

## 1. Introduction

West Nile Virus (WNV) is a member of the *Faviviridae* viral family, originally observed in tropical areas in the 1930s [1]. Most notably, WNV is a neurotrophic mosquito-borne pathogen, which is responsible for the development of West Nile Fever disease. While the vast majority of the human infections evolve with only mild symptoms, or even in totally asymptomatic and subclinical forms, serious outcomes may be observed in a non-negligible minority of cases [2,3]. These are mostly related to the development of encephalitis, which can be fatal in humans, birds, and horses. More specifically, West Nile Fever may result in fever and myalgias, which can evolve into meningoencephalitis and death. While encephalitis appears in a limited number of patients, it can progress to a severe neurological outcome and induce acute flaccid paralysis, manifesting after meningitis or encephalitis, with rapidly progressing symptoms. In addition, severe poliomyelitis-like syndromes have been observed and related to a poor prognosis [3].

The transmission and life cycle of WNV is based on an equilibrium involving the vector, i.e., mosquitoes, and the reservoir, mainly birds [4,5]. The maintenance of the birds–mosquitoes–birds cycle and its transmission through mosquito’s bites ensure the diffusion of the virus [6], although some instances of direct bird-to-bird contamination have been reported [7]. In the same cycle, WNV may be transferred to humans, or horses, via infected mosquitoes. However, it should be noted that human contamination represents a dead end for the viral diffusion, stopping its reproduction [4].

Originally present in tropical areas, WNV progression through European and American temperate areas became evident at the end of the 20th Century when outbreaks were observed in Cyprus and in the New York City area (1999) [8]. Since then, WNV evolved seasonally and endemically in temperate western areas, including Europe, with periodic, though relatively limited, outbreaks observed regularly [2,9]. The colonization of temperate areas by tropical pathogens is a common phenomenon involving different emerging viruses and is also a side-effect of climate change and global warming, facilitating the adaptation of both the virus and the vector to northern geographical areas [10]. The absence of both a vaccine or antiviral targeting WNV further complicates its management and the containment of the infection [11].

From a microbiological point of view, WNV belongs to the flaviviruses genus [4,6,12], which includes other emerging pathogens such as Dengue, Zika, and Yellow Fever. Other members of the *Faviviridae* family include the hepacivirus genus [13], which is constituted mainly by hepatitis viruses. As the other members of its family, WNV is a relatively small, enveloped, positive-sense RNA virus, whose genome is constituted by approximately 11,000 bases [4,14]. Upon infection, the viral genome exploits the cellular machinery to code for three structural and seven non-structural proteins; as is usual in RNA viruses, the genome expresses a polyprotein, which should be cleaved by viral and cellular proteases to ensure its maturation to a functional form. After the protein expression and RNA duplication taking place in the cytoplasm, the novel virions are assembled in the endoplasmatic reticulum (ER) and finally excreted following their maturation. Interestingly, the coding regions of the RNA are flanked at the 3${}^{\prime }$- and 5${}^{\prime }$-end by untranslated regions (3${}^{\prime }$- and 5${}^{\prime }$-UTR), which present a high density of secondary structure motifs, mainly stem-loops, and which plays a fundamental role in the virus maturation [6].

Recently, it has been recognized that the 5${}^{\prime }$-UTR region of the viral genome is also crucial in the interplay with the host immune systems and, in particular, in the recognition of viral material by the innate immune system [15,16]. This immune pathway, which is also favored by interferon production, is firstly triggered by the recognition of double-stranded RNA regions by the oligoadenylate synthetase (OAS) enzymes. The formation of the protein/nucleic acid complex activates OAS, which synthesizes short linear adenine oligonucleotides, which in turn induce the activation of the RNAse L through their dimerization [17,18,19]. In turn, the active RNAse L cleaves all the viral and cellular RNA fragments, hence stopping its expression and, consequently, the viral maturation.

In particular, we have shown that the OAS1 enzyme is able to selectively recognize the presence of the 5${}^{\prime }$-UTR RNA from SARS-CoV-2, triggering the immune response. More specifically, we revealed [20] that the recognition proceeds through a complex interaction network involving both the RNA backbone and some dangling nucleobases and involves also a non-trivial deformation of the nucleic acid duplex. Moreover, the interaction is also favored by the flexibility of the RNA stem-loop, which may assume a highly bent conformation to accommodate in the OAS1 recognition pocket. Our results, based on full-atom classical Molecular Dynamics (MD) simulations, are also echoed by the fact that single-point mutations of OAS1 modulating its co-localization with SARS-CoV-2 correlate well with severe COVID-19 [21,22].

As a matter of fact, a strong relationship between OAS epitopes, point mutations, and West Nile Fever severity has also been emphasized, both in humans and in murine models [23,24]. Indeed, it has been observed that some single-nucleotide polymorphisms, affecting the efficiency of OAS1, are correlated with a high susceptibility to the infection and a higher probability of encephalitis development [25]. The OAS/RNAse L path is highly involved both in the first phases of the infection to facilitate its control by the host and in the activation of adaptive T and B cell-mediated immune response.

Considering the important role of OAS1 and its recognition of 5${}^{\prime }$-UTR fragments in the control of WNV infection, we report a molecular modeling and simulation study to provide the structure and the conformational ensemble of the WNV 5${}^{\prime }$-UTR fragment, specifically focusing on the first stem-loop (SL1). Besides, we characterized the dynamics of the complex between the RNA fragment and OAS1. Our results allowed us to pinpoint the specific interactions leading to the recognition and the differences observed compared to other viral agents, such as SARS-CoV-2.

## 2. Materials and Methods

### 2.1. Systems’ Setup and Docking

The structure of the 5${}^{\prime }$-UTR first stem-loop of WNV RNA was generated with the RNAcomposer webserver [26], based on the sequence reported by Deo et al. [15]. The OAS1-RNA complex was built by docking the main conformation of the 5${}^{\prime }$-UTR as determined by clustering (see below) onto the OAS1 crystal structure (PDB ID 4IG8 [27]) using the HDock webserver with standard parameters (http://hdock.phys.hust.edu.cn/, accessed on 20 March 2022) [28,29]. Each system was soaked in a TIP3P water box with a 10 Å buffer, and potassium counter-ions were added to ensure global charge neutrality. The systems were then subjected to a 10,000-step minimization run.

### 2.2. Molecular Dynamics Simulations

The optimized structure of each system was taken as the starting structure for the Molecular Dynamics simulations using the NAMD3 program [30]. The AMBER ff14SB force field parameters were used to describe the system, together with the OL3 corrections for the RNA [31]. The Hydrogen Mass Repartition scheme [32] was applied to all hydrogens excluding water in order to allow, together with the SHAKE algorithm, setting the time-step to 4 fs. The system was first equilibrated in three successive 1.2 ns runs at 300 K in the isotherm and isobaric NPT ensemble, with decreasing constraints applied on the nucleic and amino acids. The Langevin thermostat with a 1 ps${}^{-1}$ collision frequency was applied, and the electrostatic interactions were treated using the Particle Mesh Ewald (PME) [33] scheme with a 9 Å cutoff. The system was then sampled without constraints along 800 ns and 500 ns for the isolated RNA and the OAS1-RNA complex, respectively.

Structural analysis of the RNA strand was performed using the Curves+ program [34], and the RMSD and distances were monitored using the cpptraj module of AMBER18. The bend angle between the two stems was computed as the angle between the centers of mass of the first stem section, the base of the large loop (residues 17–19, 46–48, and 58–60), and the second stem section. The flexibility profiles were generated using a program derived from a machine learning protocol described by Fleetwood et al. [35], following a strategy successfully used on different biomolecules [36,37,38]. Conformational ensembles were clustered with respect to the RMSD of the RNA and/or the protein using a hierarchical agglomerative method with average-linkage to calculate inter-cluster distances.

Visualization, picture rendering, and formatting were carried out using the VMD and Inkscape software [39,40]. Time series and histograms were plotted using the ggplot2 [41] package in RStudio [42].

## 3. Results

### 3.1. Structure and Dynamics of the West Nile 5${}^{\prime }$-Untranslated Region

As reported by experimental works [15,43,44], the first stem-loop of the WNV 5${}^{\prime }$-UTR is composed of 74 nucleic acids organized in two stem-loop (SL) sections at positions 4–16/61–74 (first) and 20–28/37–45 (s)—see Figure 1A. The two SLs are connected by a short 3-residue segment on one side and by a large 15-residue loop on the other side. The second section adopts an ideal helix shape terminated by a short loop (residues 29–36), while the first one harbors an extruded uracil residue at position 67.

The 3D model generated with the RNAcomposer server [26] exhibits a 74${}^{\circ }$ kink between the two stem sections. Unbiased MD simulations of the isolated 5-UTR structure showed that this angle increases upon relaxation, reaching values of 109 ± 12${}^{\circ }$—see Figure 1B. The clustering of the sampled conformational ensemble reveals the structural convergence towards one major conformation (75% of the simulation time), supported by a plateau in the RMSD—see Appendix A. In this major conformation, the large lateral loop (residues 46–60) shrinks into a stem-like structure, yet no base-pairing, hence no double-helical organization, can be observed. The two stem structures are conserved along the simulation, yet they show a rather different dynamic behavior.

In the first stem section, the rU67 residue that was suggested to be extruded from the helix is not involved in the base-pairing network, but it rapidly rotates within the major groove and strongly interacts with the adjacent rC66 nucleobase (rU67@O4-rC66@N2 distance at 3.0 ± 1.1 Å). The double-helical conformation is conserved along the trajectory, with standard structural properties—see Figure 2.

Noteworthy, the bend angle of this stem section, which is one of the most important descriptors of ds-RNA helical structures, evolves around 29 ± 13${}^{\circ }$ in the MD trajectory. Thus, while it remains in the standard range for nucleic acid helices, it also shows non-negligible fluctuations, underlining a certain degree of flexibility in the helical conformation, which might be important in the recognition by OAS1.

The second stem section exhibits a slightly more pronounced helicity, with a lower and less-fluctuating bend angle (17 ± 9${}^{\circ }$). The monotone behavior of this second section might be favored by the presence of a single mismatch in the pairing of the two strands’ sequences (rG22-rG43), while the first stem region harbors a mismatch (rU6-rG72), the extruded rU67 base, and the 5${}^{\prime }$- and 3${}^{\prime }$-ends’ nucleotides, which commonly show a higher flexibility. The short loop at the extremity of this stem section exhibits several flipped and exposed nucleotides, namely rA31, rG32, and rU33. These accessible residues could be important for the 5${}^{\prime }$-UTR recognition by OAS1, as we previously observed in the case of the SARS-CoV-2 genome [20].

The per-residue flexibility profile of the RNA sequence reveals similar values for the two stem sections, yet slightly higher values are observed for residues adjacent to the loops: rG15-rU16, rA27, rA44-rG45, rA61—see Figure 2C. Unsurprisingly, the residues forming the large lateral loop exhibit the highest flexibility values. Along the 800 ns simulation, the ideal helical structure, as built in the initial model, undergoes strong distortions, due to the lack of a stable interaction network between the nucleotides. Transient π-stacking between adjacent nucleobases is observed; however, the loop sequence globally fails to reach a persistent base-pairing and, thus, remains highly dynamic.

### 3.2. Binding Mode to OAS1

Docking of the major conformation of the 5${}^{\prime }$-UTR onto the human OAS1 crystal structure (from PDB ID 4IG8) results in a nucleic acid/protein complex exhibiting a contact surface that concerns the first stem section of the RNA and the canonical binding region of OAS1—see Figure 3A. In the initial structure, the second stem section, instead, does not interact with OAS1. Some amino acids are found nearby the backbone atoms of the RNA lateral loop, namely R210 and K14 in close contact with rA60 and rA57, respectively, yet the ds-RNA length in contact with the protein is limited to 13 bp, while some studies suggested a minimum length of 18 bp for OAS1 recognition [45]. However, upon MD simulation, we showed that the protein/RNA complex might adopt a different binding mode, involving the second stem section, rather than the lateral loop—see Figure 3A.

Indeed, the OAS1-RNA structure rapidly converges (see the RMSD evolution in Figure 3B) towards a major conformation, which persists during 82% of the simulation, as shown by the clustering analysis—see Appendix A. The second stem section rotates towards the protein, increasing the previously described RNA kink to 143 ± 7${}^{\circ }$. Interestingly, after the rotation of the 20–45 region (125 ns), both stem sections exhibit a ∼20 ± 10${}^{\circ }$ bend angle—see Figure 4A,B. A change in the bending angles of the ds-RNA upon binding to OAS1 has also been suggested for canonical ds-RNA recognition [46]. The flexibility profile of the RNA is also perturbed, with a much shallower trend—see Figure 4C. As for the isolated RNA system, peak values are found for nucleic acids neighboring the loops, yet much less pronounced. Likewise, the fluctuations of the large 46–60 loop are strongly reduced upon binding to the protein, the overall structure being much more constrained by the double interaction between OAS1 and the RNA stem regions. As a matter of fact, without being in contact with the protein and contrary to what is observed in the isolated RNA system, this loop exhibits a stable interaction network. Persistent hydrogen bonds are found between rU49/rA50 and the backbone of the facing rC55 and between the rA51 and rA54 nucleobase and backbone atoms—see Appendix A.

At the OAS1-RNA contact surface, sequence, RNA, and non-specific interactions can be pinpointed. As reported in the case of SARS-CoV-2 genome recognition [20], the main interactions take place between the amino acids and the RNA backbone.

More specifically, an extensive hydrogen bonds network is observed between the first stem section, mainly featuring R27, N31, K60, R195, T203, and K206, and the nucleic phosphates of rG17 (3.0 ± 1.7 Å), rU16 (1.9 ± 0.4 Å), rG15 (2.9 ± 0.7 Å), rC11 (2.0 ± 0.2 Å), rU59 (2.8 ± 1.4 Å), and rA60 (3.2 ± 0.8 Å), respectively—see Figure 5. Such non-specific interactions are also found between the second stem region and the protein: rG37 with the 5${}^{\prime }$ terminal M1 and rU39 with S11. 

Furthermore, we may also observe an RNA-specific recognition of the ribose 2${}^{\prime }$-hydroxyl groups, which involves different protein residues for the two stem sections. On the first SL, interactions are formed between D35 and rU14 (4.0 ± 1.3 Å), K42 and rC11 (3.4 ± 1.0 Å), E43 and the extruded rU67 (3.6 ± 1.3 Å), and T160 and rG9 (3.3 ± 1.6 Å). Only one of such ribose-specific interactions can be identified with the second RNA stem region, which involves N6 and rG37 (3.1 ± 1.0 Å).

While it has been suggested that the recognition of exogenous genetic material by OAS1 does not rely on the Watson–Crick base network [45], our simulations capture few sequence-specific interactions between OAS1 and the WNV 5${}^{\prime }$-UTR nucleobases. Indeed, a very persistent hydrogen bond is observed between Q158 and the rG9 base (2.0 ± 0.6 Å). Interestingly, the proximal G157 has been proposed to be crucial for the specific binding of OAS1 to ds-RNA [45,46]. Yet, its effect might be due to the fact that it is a direct neighbor of Q158, which is known to interact with the RNA backbone and nucleobases. Besides, K42 is also prone to interact with the RNA nucleobase of the extruded rU67 (3.5 ± 1.1 Å), which is a an important structural feature inherent to the WNV 5${}^{\prime }$-UTR first SL.

## 4. Discussion and Conclusions

The recent increase in WNV outbreaks in Europe [9] makes it an emerging virus, which should be kept under surveillance. Here, we resorted to all-atom MD simulations in order to unravel the molecular mechanisms driving the recognition of the WNV genome by the human immune system. We especially focused on the interaction between the first stem-loop of the viral 5${}^{\prime }$-UTR and the human 2${}^{\prime }$-5${}^{\prime }$ OAS1 protein that triggers the innate immune response through the OAS1/RNAse L pathway.

Simulations of the isolated RNA structure showed the good stability of the hypothesized secondary structure. The dynamic behavior of the 5${}^{\prime }$-UTR exhibited a convergence towards a major conformation with a kink of 109 ± 12${}^{\circ }$ in the stem-loop caused by the presence of the lateral disordered loop dividing the stem-loop into two separated helical sections. No stable interaction could be found in the large loop spanning residues rG46 to rA60, which remains disordered, yet shrinks into a hairpin-like shape. Importantly, the bend angle of the first section exhibits non-negligible fluctuations that might be important for the recognition by OAS1. Indeed, this section harbors the WWNNNNNNNNNWG motif (W = A or U; N = non-specific) invoked in the literature [47] as a consensus sequence at both its 5${}^{\prime }$- and 3${}^{\prime }$-ends. The prevalence of this motif at the surface contact with OAS1 was also confirmed by the docking results—see Figure 1 and Figure 4.

MD simulations of the OAS1/5${}^{\prime }$-UTR complex highlight the potential of OAS1 to stabilize the RNA structure upon binding, with a major binding mode involving contacts with the two stem sections. The formation of the complex results in an increase of 35${}^{\circ }$ of the RNA SL kink, inducing a geometrical constraint on the lateral loop, which then maintains a stable conformation. An extensive interaction network appears, which provides a high stability to the complex, which consequently induces a drop of the flexibility of the RNA structure—see Figure 2C and Figure 4C. The key identified amino acids interacting with the first stem section are in good agreement with previous studies concerning ds-RNA recognition by OAS1 [27,46]. However, the contact interface with the second stem region has not been characterized for other ds-RNA, suggesting a specific binding mode for the WNV 5${}^{\prime }$-UTR that allows compensating for the short length of the first stem section (13-bp), which might not be sufficient to enforce an efficient recognition by OAS1. Indeed, it has been reported that a minimum of 18 bp-length ds-RNA is necessary to ensure the formation of stable complexes [45]. On the 5${}^{\prime }$-end, the rG15 matches the nucleic acid consensus motif (${}^{3}$UAAUUCGCCUGUG${}^{15}$) and interacts with K60, yet rG9 seems to be even more important for specific binding through its sugar ring (with T160) and its nucleobase (Q158). Likewise, the ds-RNA consensus motif at the 3${}^{\prime }$-end should involve rG72 (${}^{60}$AACACAGUGCGAG${}^{72}$), yet the extruded rU67 nucleotide seems to be crucial for binding to OAS1 in the case of the WNV genome, as it interacts not only through its sugar ring with E43, but also specifically with K42 through its nucleobase—see Figure 5.

Overall, our molecular dynamics simulations revealed an atypical binding mode of the first stem-loop of the WNV 5${}^{\prime }$-UTR to OAS1, featuring an additional contact interface with the second helical section of the RNA. We provided a mapping of the OAS1 amino acids involved in the recognition of this type of RNA conformation, which is extended compared to the canonical ds-RNA as found in the crystal structure [27] or the SARS-CoV-2 5${}^{\prime }$-UTR that we previously investigated [20]. In the future, we plan to extend our study to include the interaction of different OAS isoforms such as OAS2 and OAS3 with viral, and specifically WNV, 5${}^{\prime }$-UTR RNA. This will also allow providing a global vision of the immune system recognition of exogenous RNA material. Our results bring new insights into the molecular mechanisms behind the specific recognition of the viral genome by the OAS1 protein, the polymorphisms of which have been shown to be linked to several diseases’ severity. However, care should be taken when considering immune system specificity. Indeed, the efficacy of the immune system is usually linked to the recognition of sequence or structural features belonging to a large spectrum of viruses, thus offering adequate protection. In this respect, in the future, we also plan to perform combined bio-informatic and molecular modeling and simulation studies, also based on the present results, to identify key structural patterns triggering OAS1 response involving different families of RNA viruses. Thus, our results could also foster the development of novel antivirals, especially in the framework of RNA therapeutics [48,49] for emerging viral threats.

## Figures and Tables

**Figure 1 viruses-14-01282-f001:**
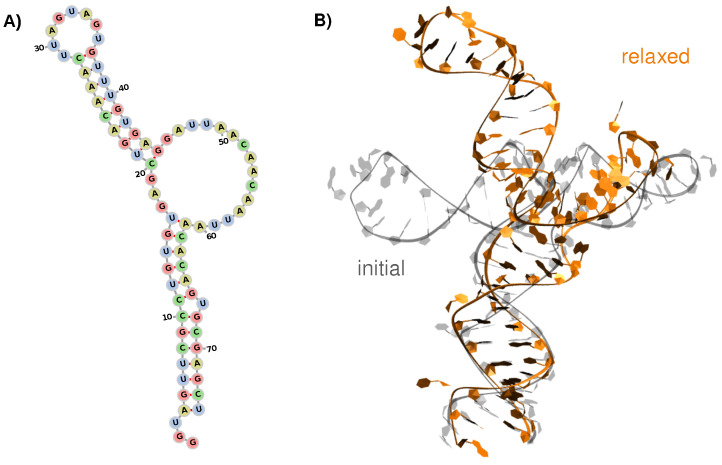
(**A**) Secondary and (**B**) tertiary structure of the first stem-loop of the WNV 5${}^{\prime }$-untranslated region. The Secondary structure is colored by residue type. The tertiary structure shows both the initial model (transparent grey) and the major conformation obtained by MD simulations (orange).

**Figure 2 viruses-14-01282-f002:**
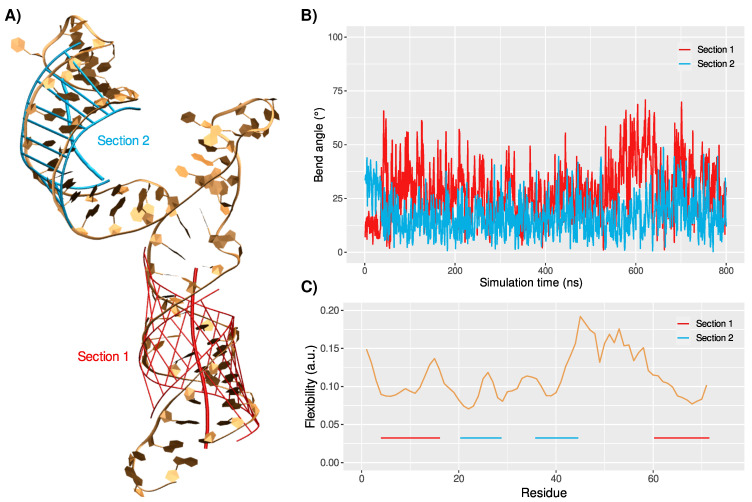
(**A**) Projection of the bend and network interaction in the stem sections 1 (red) and 2 (cyan) onto the 5${}^{\prime }$-UTR RNA major conformation. (**B**) Evolution of the local bend angles of the two stem sections. (**C**) Flexibility profile per residue of the 5${}^{\prime }$-UTR RNA. The stem regions are indicated by red (first section) and cyan (second section) lines.

**Figure 3 viruses-14-01282-f003:**
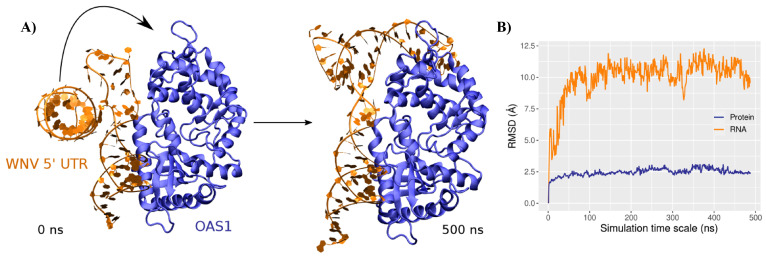
(**A**) Structure of the initial RNA-OAS1 complex generated by docking (**left**) and of the major conformation sampled along the MD run (**right**). The WNV 5${}^{\prime }$-UTR and the OAS1 protein are depicted in orange and blue, respectively. (**B**) Evolution of the RNA (orange) and protein (blue) root-mean-squared deviation along the MD simulation.

**Figure 4 viruses-14-01282-f004:**
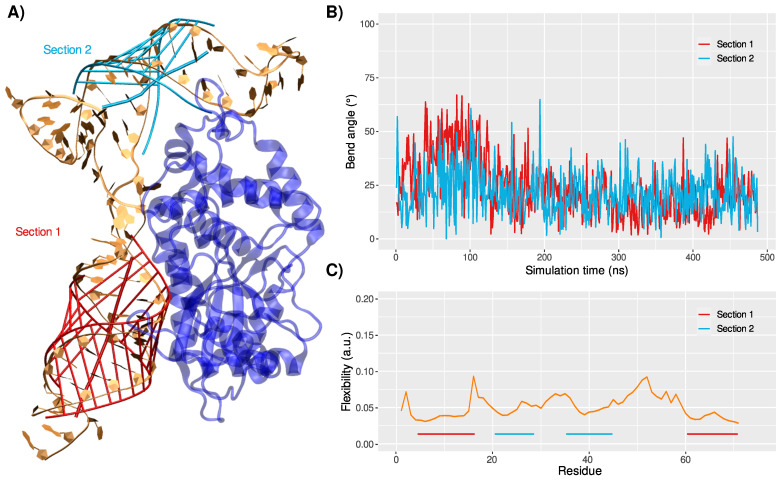
(**A**) Projection of the bend and network interaction in the stem sections 1 (red) and 2 (cyan) onto the major conformation characterized for the OAS1-RNA complex. (**B**) Evolution of the local bend angles of the two stem sections. (**C**) Flexibility profile per residue of the 5${}^{\prime }$-UTR RNA. The stem regions are indicated by red (first section) and cyan (second section) lines.

**Figure 5 viruses-14-01282-f005:**
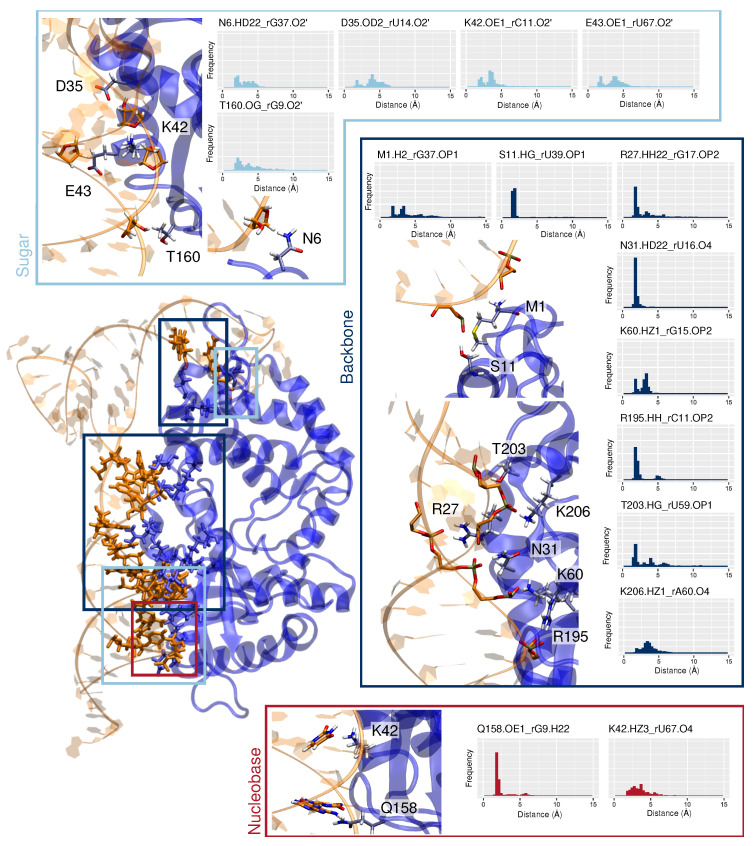
Representation and frequencies of the major interactions between OAS1 residues and the sugar rings (RNA-specific, light blue), backbone (non-specific, dark blue), or nucleobases (sequence-specific, red).

## Data Availability

Not applicable.

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
