# Peer review of "Specific Recognition of the 5′-Untranslated Region of West Nile Virus Genome by Human Innate Immune System"

_viruses, 2022, doi:10.3390/v14061282_

Round 1

Reviewer 1 Report

Bignon et al utilizes molecular dynamics simulations tool to characterize the 5` UTR of West Nile Virus (WNV) genome and its recognition by human oligoadenylate synthetase (OAS) in this manuscript. The same group earlier performed such simulations study in SARS-CoV-2 genome and OAS1. Given the possibility that innate immune response triggers differently with viral genetic material, it is of interest to identify the dynamics and interaction pattern between virus and host protein. Bignon et al identified some key differences such as ds-RNA length is limited to 13 base pair in order to contact with the protein. Further, they show that a different binding mode involving the second stem section exist for protein/RNA complex. Overall, this manuscript suggested an atypical binding mode is present in the WNV 5`UTR by the first stem loop and an additional second helical section of the RNA. This further understands the mechanism of WNV genome recognition by the OAS1 protein.

This is well articulated study, and the presented data supports the conclusion. This manuscript is well written and provides interesting details in the field host-virus interaction, particularly recognition of viral genome and activation of host immune responses.

Minor comments:

  1. At line 16 and 46, there are typos – “faviridae”
  2. Figure 3 (B) – time scale (ns) should be added for x axis

Author Response

Bignon et al utilizes molecular dynamics simulations tool to characterize the 5` UTR of West Nile Virus (WNV) genome and its recognition by human oligoadenylate synthetase (OAS) in this manuscript. The same group earlier performed such simulations study in SARS-CoV-2 genome and OAS1. Given the possibility that innate immune response triggers differently with viral genetic material, it is of interest to identify the dynamics and interaction pattern between virus and host protein. Bignon et al identified some key differences such as ds-RNA length is limited to 13 base pair in order to contact with the protein. Further, they show that a different binding mode involving the second stem section exist for protein/RNA complex. Overall, this manuscript suggested an atypical binding mode is present in the WNV 5`UTR by the first stem loop and an additional second helical section of the RNA. This further understands the mechanism of WNV genome recognition by the OAS1 protein.

This is well articulated study, and the presented data supports the conclusion. This manuscript is well written and provides interesting details in the field host-virus interaction, particularly recognition of viral genome and activation of host immune responses.

We thank the reviewer for these very positive comments.

Minor comments:

  1. At line 16 and 46, there are typos – “faviridae”

This has been corrected.

  1. Figure 3 (B) – time scale (ns) should be added for x axis

We thank the reviewer for pointing out this oversight. This was corrected.

Reviewer 2 Report

The manuscript by Bignon et al. describes modeling of the 5'-UTR of West Nile virus (WNV) and potential interactions with one isoform of oligoadenylate synthetase (OAS), an antiviral component of the human innate immune response. The authors' general approach is scientifically sound, and their results are interesting. Molecular dynamics simulations of the WNV 5'-UTR reveal both its overall shape and degree of flexibility, as well as the particular positions most likely to adopt multiple conformations. Docking the RNA structure to the crystal structure of OAS1, followed by further molecular dynamics simulations, revealed an intriguing potential mode of RNA interaction in which the two stems of the WNV 5'-UTR interact with two distinct binding surfaces. To my knowledge, this is a novel mode of RNA recognition for OAS proteins with several interesting implications. 

My concerns about the manuscript stem mostly from the authors description and interpretation of the interactions between OAS and the viral RNA. In particular, it is not clear why OAS1 is assumed to be the only isoform that would be likely to interact with the WNV 5'-UTR. While there is some evidence from mouse models (Mashimo et al. 2002, reference 23 in this paper) that OAS1 is involved in WNV suppression, that work also suggests a role for other OAS isoforms. While OAS1 appears to be the only isoform with an available crystal structure, it would be worthwhile to use models (such as those generated by AlphaFold for example) of OAS2 and OAS3 to determine they exhibit similar RNA binding characteristics with the WNV genome. A somewhat similar area of concern is the authors' description of the WNV 5'-UTR/OAS1 interaction as "specific". My understanding of the innate response is that the components involved, including the OAS system, recognize features that are general to a range of pathogenic (in this case viral) targets. Thus, the details of the interactions between the 5'-UTR and OAS1 are interesting not just because of their specificity, but also in the degree to which they represent features that are general to a larger set of viral RNAs that might be recognized by OAS. I would like to see the authors address this question to a greater extent. 

Finally, the manuscript could use a bit of English language editing. The use of two particular idioms stand out: 

• On line 79, the authors use "stressed out" when I believe they mean "emphasized".

• On line 118, the authors use "in-house brewed" to describe software or scripts they have written to generate flexibility profiles. It would be more clear to simply write "our own" to describe that software.  

Author Response

The manuscript by Bignon et al. describes modeling of the 5'-UTR of West Nile virus (WNV) and potential interactions with one isoform of oligoadenylate synthetase (OAS), an antiviral component of the human innate immune response. The authors' general approach is scientifically sound, and their results are interesting. Molecular dynamics simulations of the WNV 5'-UTR reveal both its overall shape and degree of flexibility, as well as the particular positions most likely to adopt multiple conformations. Docking the RNA structure to the crystal structure of OAS1, followed by further molecular dynamics simulations, revealed an intriguing potential mode of RNA interaction in which the two stems of the WNV 5'-UTR interact with two distinct binding surfaces. To my knowledge, this is a novel mode of RNA recognition for OAS proteins with several interesting implications.

We thank the reviewer for underlining the interest of our study. 

My concerns about the manuscript stem mostly from the authors description and interpretation of the interactions between OAS and the viral RNA. In particular, it is not clear why OAS1 is assumed to be the only isoform that would be likely to interact with the WNV 5'-UTR. While there is some evidence from mouse models (Mashimo et al. 2002, reference 23 in this paper) that OAS1 is involved in WNV suppression, that work also suggests a role for other OAS isoforms. While OAS1 appears to be the only isoform with an available crystal structure, it would be worthwhile to use models (such as those generated by AlphaFold for example) of OAS2 and OAS3 to determine they exhibit similar RNA binding characteristics with the WNV genome. A somewhat similar area of concern is the authors' description of the WNV 5'-UTR/OAS1 interaction as "specific". My understanding of the innate response is that the components involved, including the OAS system, recognize features that are general to a range of pathogenic (in this case viral) targets. Thus, the details of the interactions between the 5'-UTR and OAS1 are interesting not just because of their specificity, but also in the degree to which they represent features that are general to a larger set of viral RNAs that might be recognized by OAS. I would like to see the authors address this question to a greater extent. 

We thank the reviewer for raising these concerns. 

We fully agree that investigating how other OAS isoforms would bind West Nile RNA could add to the general comprehension of the recognition of viral RNA by innate immune systems. However, such a study would require repeating all the computational analysis for the other protein systems, hence it clearly falls outside of the scope of the present contribution. Nonetheless, the recognition of the isoform is an important aspect which will make it the subject of a forthcoming contribution. This apsect is now cited in the Conclusion section.  

Concerning the specific or general features of OAS1 binding mode to viral RNA, we also agree with the reviewer on its importance. We have added a further paragraph to better specify this issue in the MS.

Finally, the manuscript could use a bit of English language editing. The use of two particular idioms stand out: 

  • On line 79, the authors use "stressed out" when I believe they mean "emphasized".
  • On line 118, the authors use "in-house brewed" to describe software or scripts they have written to generate flexibility profiles. It would be more clear to simply write "our own" to describe that software. 

This has been modified. The text has been carefully proof-read and typos have been corrected.